# Exercise-induced response of proteinogenic and non-proteinogenic plasma free amino acids is sport-specific: A comparison of sprint and endurance athletes

Krzysztof Kusy[1]*, Jan Matysiak[2], Zenon J. Kokot[3], Monika Ciekot-Sołtysiak[1], Agnieszka Klupczyńska-Gabryszak[2], Ewa Anna Zarębska[1], Szymon Plewa[2], Paweł Dereziński[2], Jacek Zieliński[1]

1 Department of Athletics Strength and Conditioning, Poznan University of Physical Education, Poznań, Poland, 2 Department of Inorganic and Analytical Chemistry, Poznan University of Medical Sciences, Poznań, Poland, 3 Faculty of Health Sciences, Calisia University, Kalisz, Poland

* kusy@awf.poznan.pl

**Data Availability Statement:** The data relevant to this paper are available in the RepOD repository at

## Abstract

Circulating blood is an important plasma free amino acids (PFAAs) reservoir and a pivotal link between metabolic pathways. No comparisons are available between athletes with opposite training adaptations that include a broader spectrum of both proteinogenic and non-proteinogenic amino acids, and that take into account skeletal muscle mass. We hypothesized that the levels of the exercise-induced PFAAs concentration are related to the type of training-related metabolic adaptation. We compared highly trained endurance athletes (n = 11) and sprinters (n = 10) aged 20–35 years who performed incremental exercise until exhaustion. Venous blood was collected before and during the test and 30-min recovery (12 samples). Forty-two PFAAs were assayed using LC-ESI-MS/MS technique. Skeletal muscle mass was estimated using dual X-ray absorptiometry method. Glutamine and alanine were dominant PFAAs throughout the whole exercise and recovery period (~350–650 µmol·L$^{-1}$). Total, combined proteinogenic, non-essential, and non-proteinogenic PFAAs levels were significantly higher in endurance athletes than sprinters (ANOVA group effects: p = 0.007, $\eta^2$ = 0.321; p = 0.011, $\eta^2$ = 0.294; p = 0.003, $\eta^2$ = 0.376; p = 0.001, $\eta^2$ = 0.471, respectively). The exercise response was more pronounced in endurance athletes, especially for non-proteinogenic PFAAs (ANOVA interaction effect: p = 0.038, $\eta^2$ = 0.123). Significant between-group differences were observed for 19 of 33 PFAAs detected, including 4 essential, 7 non-essential, and 8 non-proteinogenic ones. We demonstrated that the PFAAs response to incremental aerobic exercise is associated with the type of training-related metabolic adaptation. A greater turnover and availability of circulating PFAAs for skeletal muscles and other body tissues is observed in endurance- than in sprint-trained individuals. Non-proteinogenic PFAAs, despite low concentrations, also respond to exercise loads, indicating their important, though less understood role in exercise metabolism. Our study provides additional insight into the exercise-induced physiological response of PFAAs, and may also provide a rationale in discussions regarding dietary amino acid requirements in high-performance athletes with respect to sports specialization.

https://doi.org/10.18150/CROEEQ (doi:10.18150/CROEEQ).

**Funding:** This research project was funded by National Science Centre Poland, grant OPUS 14 No. 2017/27/B/NZ7/02828. https://www.ncn.gov.pl/ Krzysztof Kusy (KK) was the grant recipient The founder did not play any role in the study design, data collection and analysis, decision to publish, or preparation of the manuscript.

**Competing interests:** The authors have declared that no competing interests exist.

## Introduction

Amino acids, released and extracted by individual tissues, comprise the amino acid pool and are found in various organs and fluids of the human body including skeletal muscles, being their major depot. Circulating blood plasma is a temporary and small but important reservoir of free amino acids (~1% of the total pool), constituting a pivotal link between metabolic pathways, the content of which changes with exposure to stress, such as exercise, starvation, or disease [1, 2]. The movement of this fraction of the amino acid pool in the bloodstream is a ceaseless process. It is well known that with exercise intensity, skeletal muscle blood flow increases dramatically, and with it the rate and volume of amino acids flow. During exercise, the circulatory system is essentially the only fast and efficient channel for the exchange of amino acids between skeletal muscles and other tissues and organs. This allows the efficiency of metabolic processes to be continuously adjusted to meet the demands of physical exertion. Acute graded exercise until exhaustion is a potent factor in enhancing amino acid metabolism, the products of which contribute to energy supply and fatigue resistance in many interrelated pathways [3–6]. This is facilitated by the suppression of protein synthesis during exercise and the resulting availability of free amino acids – being a kind of metabolic "circulating currency" – for more stressed pathways [7].

The profile of plasma free amino acid (PFAA) concentration may reflect amino acid turnover and "utilization strategy" during exercise, depending on the metabolic adaptation related to the sports discipline. Both muscle and plasma amino acid concentration are reported to be higher in endurance athletes than in untrained individuals [7–10]. Differences in PFAA concentration between highly trained athletes representing different sports are not known, but are to be expected due to the specific demands of training and competition that lead to distinctive metabolic adaptations. Stimuli triggered by different exercise modes produce divergent responses, for example, resistance exercise promotes myofibrillar, while endurance mode mitochondrial protein synthesis in trained individuals [11, 12]. However, direct comparisons of exercise-induced PFAA concentrations between groups of specialized high-level athletes are lacking.

There are many studies on the metabolism of proteinogenic amino acids in various body tissues and fluids. In contrast, the issue of exercise-induced levels of non-proteinogenic PFAAs is poorly explored, with only one recent study known to consider a few non-proteinogenic PFAAs in untrained individuals [13]. Non-proteinogenic amino acids are not encoded by the genome or incorporated into proteins during translation. However, they serve as intermediates in biosynthesis and perform various physiological functions during exercise [14]. Thus, the inclusion of non-proteinogenic PFAAs seems to be noteworthy in the context of the metabolic response to exercise.

To our knowledge, previous studies have not made direct comparisons of proteinogenic or non-proteinogenic PFAA concentrations between individuals with opposite metabolic adaptations to athletic training. In addition, multiple blood sampling is not the rule, so the literature mainly provides pre- and post-exercise data, ignoring the course of PFAA changes during progressive exercise. We hypothesize that the magnitude and time course of the exercise-induced PFAA concentrations, both proteinogenic and non-proteinogenic, depend on the type of training-related metabolic adaptation. For this purpose, we compare highly trained endurance athletes and sprinters. The results of our study may provide additional insight into the exercise-induced physiological response of PFAAs, and may also serve as a rationale in discussions regarding dietary amino acid requirements in high-performance athletes with respect to sports specialization.

## Methods

### Participants

Eleven endurance (long-distance runners and triathletes) and 10 sprint-trained (sprint runners) athletes, aged 25.3±5.0 and 25.1±3.3 years, respectively, participated in the study. The inclusion criteria were (age 18–35 years, (ii) involvement in a planned and structured training process for at least 5 years, (iii) at least a national or international sports level, (iv) being a licensed athlete eligible to compete, as certified by the relevant sports association, (v) undergoing periodic medical examinations recommended by the relevant sports association, (vi) being in good health and free of injuries during the testing period, and (vii) lack of a positive anti-doping control result. The study was approved by the Ethics Committee at the Poznan University of Medical Sciences (decision No 1252/18 issued on 6 December 2018) and performed according to the ethical standards as laid down in the 1964 Declaration of Helsinki and its later amendments. Athletes were recruited from competitive sports clubs and national teams between January 1 2019 and September 30 2021. The participants gave their written informed consent to participate before entering the study. The project was retrospectively registered in ClinicalTrials.gov ID: NCT05672758, released on January 5, 2023. The data utilized are available in the open RepOD repository (https://doi.org/10.18150/CROEEQ).

### Study design

Our goal was to compare exercise-induced PFAA levels in two highly trained athletic groups of opposite specializations. We expected differences in the PFAAs concentration due to fixed training-related metabolic adaptations. The essential part of this study was a graded exercise test until exhaustion. Before, during, and after the test, blood samples were taken to picture the change in PFAA concentrations (primary outcomes) as affected by exercise intensity and the recovery phase. A series of secondary variables were also determined for description and data interpretation. The measurements were performed in the pre-competition period of a one-year training cycle, i.e. when athletes approached the highest level of physiological adaptation specific to their disciplines. Exercise testing, body composition measurements, and blood draws were conducted at the Human Movement Laboratory "LaBthletics" of the Poznan University of Physical Education, Poznan, Poland. Plasma samples for PFAAs assay were analyzed in the Department of Inorganic and Analytical Chemistry of the Poznan University of Medical Sciences, Poznan, Poland.

### Dietary control

In the one-month period prior to laboratory testing, customary diet was assessed by recording the foodstuffs, dishes, beverages, and dietary supplements (including protein and amino acids) consumed during a typical week covering weekdays and weekends. To estimate the amount of food and beverages consumed at each meal, participants used household measures or grams and verified them using a specially developed photo album of foods and dishes [15]. The energy and nutrient intake was then analyzed and calculated by a sports dietitian using Diete-tyk-2 software (JuMaR, Poznań, Poland). Complete nutritional data, ensuring a valid analysis, were eventually obtained from 7 (out of 11) endurance athletes and from 6 (out of 10) sprinters. On the day of the measurements, athletes arrived at the laboratory at 7 a.m. after an overnight fast. Prior to that, they were asked not to modify their dietary habits in the days before the visit and not to hydrate extra or dehydrate before the test. No supplements or aids potentially affecting PFAA levels and physical performance were allowed. In addition, participants briefly reported on their food and supplementary intake over the past 48 hours to detect

unusual eating behaviors that could affect the study results (in which case laboratory measurements could be rescheduled).

## Skeletal muscle mass estimation

Height and weight were measured using the SECA 285 station (SECA GmbH, Hamburg, Germany). Details of body composition measurements were described elsewhere [16]. In brief, the dual X-ray absorptiometry method (Lunar Prodigy device and enCORE 16 SP1 software; GE Healthcare, Chicago, USA) was applied. The body components of interest were appendicular lean body mass (ALBM) and fat mass (FM). Based on appendicular lean body mass, total skeletal muscle mass (SMM) was calculated using the regression equation proposed by Kim et al. [17] who provided accurate and reliable estimates of total-body SMM in athletes compared to the magnetic resonance imaging method ($R^2$ = 0.98–0.99, SEE = 1.06–1.46 kg).

## Exercise protocol

The tests were conducted between 8 and 11 a.m. The details of the exercise protocol were described elsewhere [16]. In short, participants did not engage in high-intensity and long-duration training sessions 24–48 hours before the laboratory visit. The ambient temperature remained unchanged at 20–21°C. Participants performed an incremental running test on the h/p Cosmos Pulsar treadmill (Sports & Medical GmbH, Nussdorf-Traunstein, Germany). Initially, the athlete stood still on the treadmill for 3 min, then the speed was set at 4 km·h$^{-1}$ and after another 3 min increased to 8 km·h$^{-1}$. After that, the speed was progressively increased by 2 km·h$^{-1}$ every 3 min until voluntary exhaustion. Total exercise duration ranged from 20 min (sprinters) to 24 min (endurance athletes). Respiratory variables were measured with the MetaLyzer 3B ergospirometer and processed with the MetaSoft Studio 5.1.0 software package (Cortex Biophysik GmbH, Leipzig, Germany). Heart rate was measured with the Polar Bluetooth Smart H6 monitor (Polar Electro Oy, Kempele, Finland). The main variable of interest was maximal oxygen uptake ($\dot{V}O_2$max).

## Blood sampling

A peripheral venous catheter was placed into the antecubital vein. Blood samples were drawn before exercise, five times during exercise (every 3 min, beginning at 10 km·h$^{-1}$ and ending at exhaustion), and five times during post-exercise recovery (5, 10, 15, 20, and 30 min after exercise completion), up to a total of 12 samples (each 2.5 ml) depending on the maximum speed. The samples were collected into plasma-separation tubes (containing EDTA) and centrifuged at 13,000 rpm for 3 min at 4°C (Universal device, Hettich GmbH, Tuttlingen, Germany). Obtained plasma was then pipetted into 0.5 ml vials and immediately frozen in liquid nitrogen. The samples were then stored at -80°C until analysis.

## Plasma free amino acids assay

Forty-two PFAAs were assayed, among them 20 proteinogenic: L-histidine (His), L-isoleucine (Ile), L-leucine (Leu), L-lysine (Lys), L-methionine (Met), L-phenylalanine (Phe), L-threonine (Thr), L-tryptophan (Trp), L-valine (Val), L-arginine (Arg), L-cystine (Cyss), L-glutamine (Gln), glycine (Gly), L-proline (Pro), L-tyrosine (Tyr), L-alanine (Ala), L-asparagine (Asn), L-aspartic acid (Asp), L-glutamic acid (Glu), L-serine (Ser); and 22 non-proteinogenic: 1-methyl-L-histidine (1MHis), 3-methyl-L-histidine (3MHis), L-α-amino-adipic acid (Aad), L-α-amino-n-butyric acid (Abu), D,L-β-aminoisobutyric acid (bAib), β-alanine (bAla), L-citrulline (Cit), ethanolamine (EtN), hydroxy-L-proline (Hyp), L-ornithine (Orn), O-phospho-

ethanolamine (PEtN), sarcosine (Sar), taurine (Tau), L-argininosuccinic acid (Asa), cystathionine (Cth), L-anserine (Ans), L-carnosine (Car), L-homocitrulline (Hcit), L-homocystine (Hcy), O-phospho-L-serine (PSer), γ-amino-n-butyric acid (GABA), δ-hydroxylysine (Hyl).

The analyses were performed using a well-established and validated method, which was based on the Liquid Chromatography Electrospray Ionization Tandem Mass Spectrometry (LC-ESI-MS/MS) technique and the aTRAQ™ (Sciex, Framingham, MA, USA) reagent. A detailed description and validation of the method used have been presented elsewhere, and its high specificity, accuracy, and precision in determining PFAA concentrations have been confirmed [18]. The lower limit of quantitation for each AA is listed in the S1 Table. Before each sequence, the system suitability test was conducted to check retention time repeatability, the intensity of the internal standard peaks (instrument sensitivity), and the overall LC-ESI-MS/MS performance. To avoid biased results, samples were prepared and analyzed in a random order. All PFAA concentrations during exercise and recovery were corrected for changes in plasma volume relative to pre-exercise values [19]. The concentrations were then expressed per kg of total SMM to show the turnover and availability of circulating PFAAs relative to the most perfused tissue during exercise.

The following protocol for the preparation of plasma samples for amino acid assay was used. A 40-μL sample was transferred to an Eppendorf tube. To precipitate proteins present in plasma, 10 μL of 10% sulfosalicylic acid was added, then the content was mixed and centrifuged (10,000 g for 2 min). After that, the supernatant was transferred to a new tube and mixed with 40 μL of borate buffer. A 10-μL aliquot of the obtained solution was subsequently labeled with the aTRAQ reagent Δ8 solution (5 μL), mixed, centrifuged, and incubated at room temperature for 30 minutes. The labeling reaction was then stopped by the addition of 5 μL of 1.2% hydroxylamine, mixing, and incubation at room temperature for 15 minutes. After that, 32 μL of the internal standard solution was added and the content was mixed. The internal standard solution contained the same amino acids labeled with the aTRAQ reagent Δ0. Thus, each determined amino acid had its corresponding internal standard. In the next step, the sample was evaporated in a vacuum concentrator (miVac Duo Concentrator, Genevac, Stone Ridge, NY, USA) for 15 min to reduce the volume to ~20 μL. The residue was then diluted with 20 μL of water, mixed, and transferred to an autosampler vial with an insert. Norleucine and norvaline, two non-proteinogenic amino acids, were used to evaluate the labeling efficiency and recovery. Their corresponding internal standards were also present in the internal standard solution.

The analyses were performed using the liquid chromatography instrument 1260 Infinity (Agilent Technologies, Santa Clara, CA, USA) coupled with the 4000 QTRAP mass spectrometer (Sciex, Framingham, MA, USA). The mass spectrometer was equipped with an electrospray ionization source and three quadrupoles to enable quantitative analysis of the targeted amino acids. The chromatographic separation was performed with the Sciex C18 (5 μm, 4.6 mm x 150 mm) chromatography column. The flow rate of mobile phases was maintained at 800 μL/min. The method used the following mobile phases: water (phase A) and methanol (phase B), both with the addition of 0.1% formic acid and 0.01% heptafluorobutyric acid. The time of analysis was 18 minutes and during that time the chromatographic separation was carried out with the following gradient elution: from 0 to 6th minute–from 2% to 40% of phase B, then maintained at 40% of phase B for 4 min, increased to 90% of phase B till 11th minute and held at that ratio phases for 1 min, then decreased to 2% of phase B, and finally maintained at 2% of phase B from 13th till 18th minute. The injection volume was set at 2 μL and the separation temperature at 50°C. The ion source settings were: curtain gas = 20 psig; ion spray voltage = 4500 V; ion source temperature = 600°C; ion source gas 1 = 60 psig, ion source gas 2 = 50 psig. The mass spectrometer operated in positive ionization mode and the following

parameters were applied: entrance potential = 10 V; declustering potential = 30 V; collision cell exit potential = 5 V; collision energy = 30 eV (50 eV in case of 7 compounds); collision gas was nitrogen. The amino acids were measured in scheduled multiple reaction monitoring (sMRM) mode as shown in the S1 Table. This mode ensured high specificity and sensitivity of quantitative analyses. Data acquisition and processing were performed using the Analyst 1.5 software (Sciex, Framingham, MA, USA).

## Other blood assays

Blood lactate concentration was measured using the Biosen C-line device (EKF-diagnostic GmbH, Barleben, Germany), CV<1.5% at 12 mmol·L$^{-1}$. Blood ammonia concentration was measured using the PocketChem BA device (Arkray Inc., Kyoto, Japan), CV% = 2.3 in the range of 8–285 μmmol·L$^{-1}$. Blood count (white and red blood cells, hemoglobin, hematocrit) was performed using the Sysmex XS-1000i device (Sysmex Europe, Hamburg, Germany), WBC CV<3%; RBC,Hb, Hct, MCV CV<1.5%; platelet CV<4%. Pre-exercise creatine kinase level was determined using the Reflotron Plus device (Roche Diagnostics International AG, Basel, Switzerland), accuracy ±0.5% reflectance, precision ≤0.2% reflectance. All devices were calibrated before each exercise session or test according to the manufacturers' specifications.

## Statistical analysis

The Shapiro-Wilk test revealed normal data distribution for the key variables analyzed. The values were expressed as mean and standard deviation and parametric tests were used. Between-group differences of baseline characteristics were assessed using t-tests and Hedge's $g$ as a measure of effect size (small 0.2, medium 0.5, large 0.8, very large 1.4). To test the effects of athletic group, exercise, and their interaction on PFAA concentrations, a two-way repeated measure (within-between interaction) analysis of variance (ANOVA) was applied. A minimum total sample size of 16 participants was established using G*Power software, assuming 2 participant groups, 11 sampling points, $p$-value = 0.05, statistical power = 0.8, partial $\eta^2$ = 0.14, and correction for non-sphericity $\hat{\varepsilon}$ = 0.7 [20]. Levene's test was used to check for equality of variance. The Mauchly test was used to assess sphericity, followed by the Greenhouse-Geisser correction if sphericity was violated. The Bonferroni correction was applied as a post-hoc test if the main effects or interaction showed significance. The effect size for ANOVA was assessed with partial $\eta^2$ (small 0.01, medium 0.06, large 0.14). The significance level of $p \leq 0.05$ was adopted for all analyses. All statistical analyses were performed using the Statistica 13.3 software package (TIBCO software Inc., Santa Clara, CA, USA).

## Results

### Basic characteristics

The main resting and exercise characteristics of the endurance and sprint groups are compared in Table 1. There were no significant differences in age, years of specialized training, body height, absolute fat mass, resting hematological markers, creatine kinase, maximum heart rate, and peak lactate and ammonia levels. Sprint-trained athletes were significantly heavier, and had more SMM (absolute and percentage), less fat mass (percentage), but showed considerably lower levels of $\dot{V}O_2$max (absolute and relative to body weight and SMM). These differences reflected fixed adaptations related to the type of sport practiced.

The estimated daily intake of total energy and macronutrients from diet and dietary supplements is shown in Table 2. Significant differences were found only for absolute mass and energy from protein consumed, which were greater in sprinters than in endurance athletes.

**Table 1. Descriptive characteristics and differences between the athletic groups.**

|  | Endurance | Sprint | *p*-level | Hedge's *g* |
|---|---|---|---|---|
| **Age (years)** | 25.3±5.0 | 25.1±3.3 | 0.922 | – |
| **Specialized training (years)** | 11.4±3.5 | 10.2±3.5 | 0.533 | – |
| **Height (m)** | 1.82±0.06 | 1.85±0.05 | 0.158 | – |
| **Weight (kg)** | 73.7±7.7 | 83.4±7.1 | 0.008* | 1.30 |
| **ALBM (kg)** | 28.1±3.1 | 35.0±4.6 | 0.001* | 1.80 |
| **SMM (kg)** | 32.8±3.5 | 40.7±5.2 | 0.001* | 1.80 |
| **SMM (%)** | 44.5±1.1 | 48.6±2.4 | <0.001* | 2.20 |
| **FM (kg)** | 10.1±1.9 | 8.8±1.5 | 0.101 | – |
| **FM (%)** | 13.7±1.7 | 10.7±2.3 | 0.003* | 1.48 |
| **WBC ($10^9 \cdot L^{-1}$)** | 4.92±0.90 | 5.33±1.40 | 0.427 | – |
| **RBC ($10^{12} \cdot L^{-1}$)** | 4.79±0.39 | 4.83±0.36 | 0.788 | – |
| **Hb ($g \cdot dL^{-1}$)** | 13.9±1.0 | 14.4±0.7 | 0.208 | – |
| **HCT (%)** | 42.4±2.7 | 42.1±2.4 | 0.768 | – |
| **CK ($U \cdot L^{-1}$)** | 235±138 | 319±122 | 0.159 | – |
| **$\dot{V}O_2max$ ($L \cdot min^{-1}$)** | 5.04±0.69 | 4.29±0.40 | 0.007* | 1.32 |
| **$\dot{V}O_2max$ ($mL \cdot min^{-1} \cdot$ kg weight$^{-1}$)** | 68.2±4.1 | 51.5±3.5 | <0.001* | 4.42 |
| **$\dot{V}O_2max$ ($mL \cdot min^{-1} \cdot$kg SMM$^{-1}$)** | 153.4±11.1 | 106.2±9.7 | <0.001* | 4.51 |
| **HRmax ($b \cdot min^{-1}$)** | 188±7 | 189±7 | 0.633 | – |
| **LAmax ($mmol \cdot L^{-1}$)** | 10.5±1.7 | 9.7±1.3 | 0.292 | – |
| **NH$_3$max ($\mu mol \cdot L^{-1}$)** | 60.3±11.4 | 66.2±4.3 | 0.138 | – |

ALBM, appendicular lean body mass; SMM, skeletal muscle mass; FM, fat mass; WBC, white blood cells; RBC, red blood cells; Hb, hemoglobin; HCT, hematocrit; CK, creatine kinase; $\dot{V}O_2max$, maximum oxygen uptake; HRmax, heart rate at $\dot{V}O_2max$; LAmax, lactate blood concentrations immediately after exhaustion; NH$_3$max, blood ammonia concentration immediately after exhaustion

* significantly different between the groups

However, the daily intake of protein, fat, and carbohydrate in grams per kilogram of body weight and their percentage contribution to total energy intake did not differ significantly between the subgroups.

## General PFAAs profile

The concentration of 9 out of 42 analyzed amino acids did not exceed the lower limit of quantitation. These were argininosuccinic acid, cystathionine, anserine, carnosine, homocitrulline, homocystine, phosphoserine, γ-aminobutyric acid, and δ-hydroxylysine. The profile of the detected 33 PFAAs in endurance and sprint-trained athletes can be seen in Fig 1. Two non-essential PFAAs, glutamine and alanine, were dominant throughout the whole exercise and recovery period, with absolute concentrations fluctuating within the range of ~350–650 µmol·L$^{-1}$ (~11–21% of total PFAAs concentration). The second group consisted of 4 essential (valine, lysine, threonine, and leucine) and 2 non-essential (proline and glycine) PFAAs, each falling within the concentration range of ~100–300 µmol·L$^{-1}$ (~3–10% of total PFAAs). The concentrations of other PFAAs, including all non-proteinogenic ones, fluctuated below the value of 100 µmol·L$^{-1}$ (~0.1–3% of total PFAAs). Detailed data on the absolute concentrations (mean±SD) of individual PFAAs are available in the S2 Table, S1 Fig.

**Table 2. Estimated daily macronutrients intake from diet and dietary supplements in athletic groups.**

| | Endurance (n = 7) | Sprint (n = 6) | *p*-level | Hedge's *g* |
|---|---|---|---|---|
| **Total energy intake** | | | | |
| kcal | 3641±503 | 4452±838 | 0.054 | – |
| kcal·kg$^{-1}$ | 51.5±10.4 | 52.9±7.6 | 0.794 | – |
| **Protein** | | | | |
| g | 144±24 | 187±27 | 0.013* | 1.65 |
| kcal | 576±97 | 747±110 | 0.013* | 1.65 |
| g·kg$^{-1}$ | 2.1±0.6 | 2.2±0.2 | 0.512 | – |
| % of total energy intake | 15.9±1.9 | 17.1±2.9 | 0.372 | – |
| **Fat** | | | | |
| g | 128±24 | 161±41 | 0.099 | – |
| kcal | 1152±220 | 1448±366 | 0.099 | – |
| g·kg$^{-1}$ | 1.8±0.4 | 1.9±0.5 | 0.677 | – |
| % of total energy intake | 31.6±4.0 | 32.4±5.5 | 0.782 | – |
| **Carbohydrates** | | | | |
| g | 478±78 | 564±139 | 0.187 | – |
| kcal | 1912±314 | 2257±556 | 0.187 | – |
| g·kg$^{-1}$ | 6.7±1.4 | 6.7±1.2 | 0.927 | – |
| % of total energy intake | 52.5±4.3 | 50.5±56 | 0.481 | – |

* significantly different between the groups

## PFAA concentrations relative to skeletal muscle mass

The level of total SMM-adjusted PFAAs (Fig 2) was significantly higher in endurance than in sprint-trained athletes, with a similar pattern of change during exercise and recovery in both groups, i.e., a decrease between rest and exercise, followed by a moderate increase during recovery. The levels of combined proteinogenic (Fig 3A) and non-essential (Fig 3C) PFAAs were significantly higher in endurance-trained athletes, whereas the levels of essential (Fig 3B) and branched-chain (Fig 3D) PFAAs did not differ significantly. The pattern of change was in general similar to that observed for total PFAAs concentration, with more pronounced

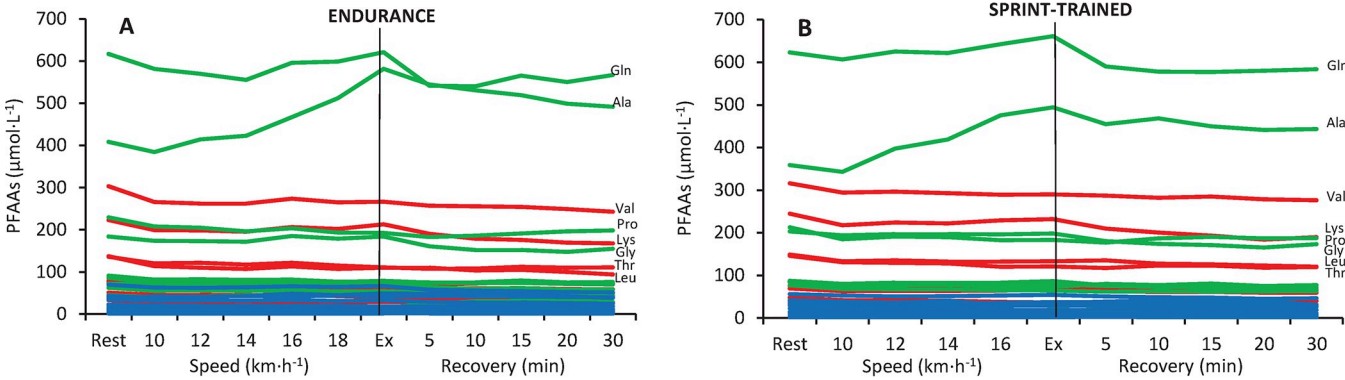

**Fig 1.** General plasma free amino acids (PFAAs) profile in endurance (panel A) and sprint-trained (panel B) athletes before exercise, during the progressive test until exhaustion (Ex), and recovery. Each line represents the course of the concentration of one amino acid (mean values). Green lines denote non-essential, red lines essential, and blue lines non-proteinogenic amino acids. Symbols for the PFAAs with the highest concentrations are given on the right: Ala, alanine; Gln, glutamine; Gly, glycine; Leu, leucine; Lys, lysine; Pro, proline; Thr, threonine; Val, valine.

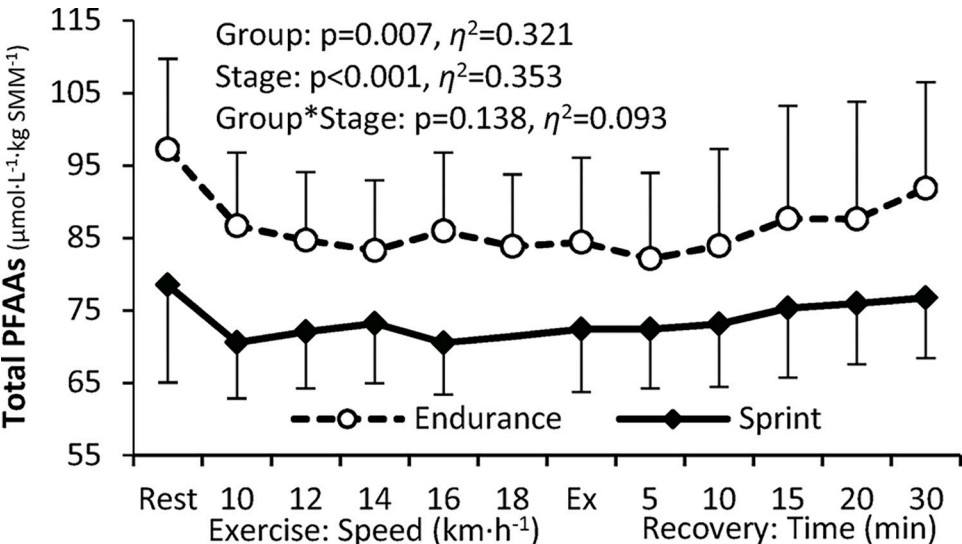

**Fig 2. Changes in the level of total plasma free amino acids (PFAAs) adjusted for skeletal muscle mass (SMM) in endurance versus sprint-trained athletes at rest, during the progressive test until exhaustion (Ex) and recovery.** ANOVA main effects for the athletic Group, Stage (of the exercise test), and Group*Stage interaction are shown (p, significance level; $\eta^2$, effect size).

changes in endurance athletes. The level of non-proteinogenic PFAAs differed significantly between the endurance and the sprint group (Fig 4), with a less pronounced change in the sprint-trained group. As far as individual PFAAs are concerned, significant between-group differences were observed for 19 of 33 PFAAs analyzed, including 4 essential (histidine,

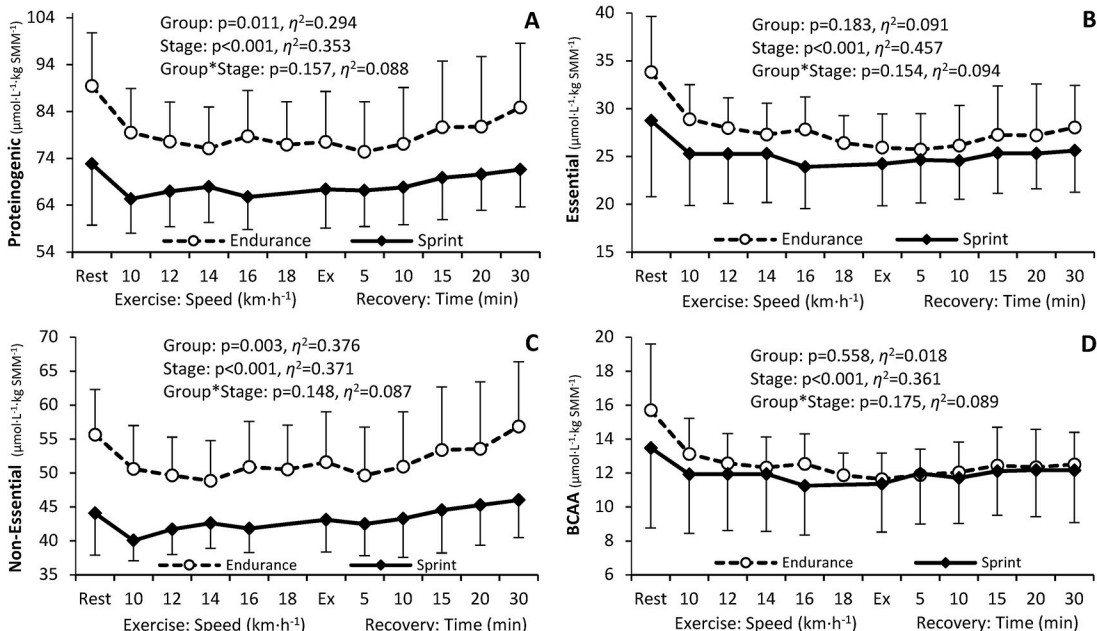

**Fig 3. Changes in the level of main groups of proteinogenic plasma free amino acids adjusted for skeletal muscle mass (SMM) in endurance and sprint-trained athletes before exercise, during the progressive test until exhaustion (Ex) and recovery.** ANOVA main effects for the athletic Group, Stage (of the exercise test), and Group*Stage interaction are shown (p, significance level; $\eta^2$, effect size).

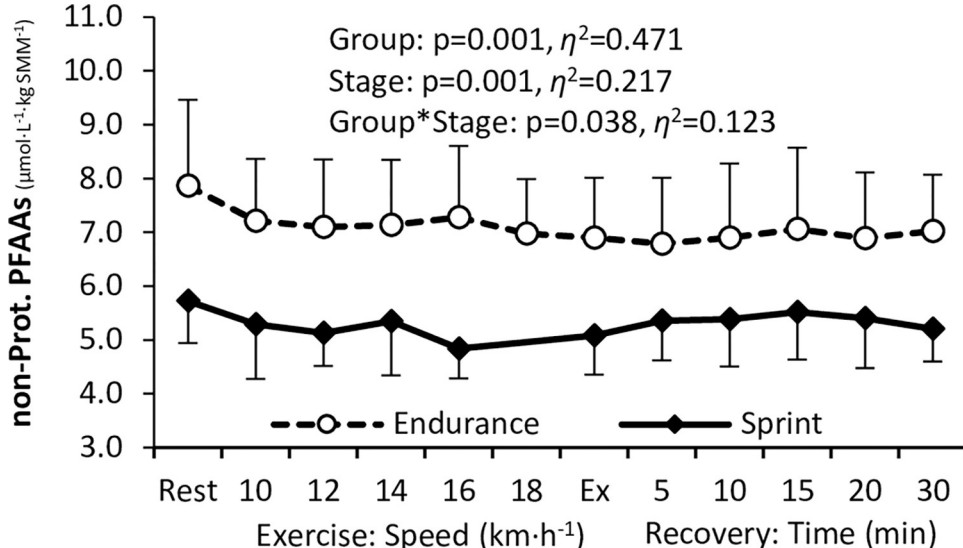

**Fig 4. Changes in the level of combined non-proteinogenic plasma free amino acids adjusted for skeletal muscle mass (SMM) in endurance and sprint-trained athletes before exercise, during the progressive test until exhaustion (Ex) and recovery.** ANOVA main effects for the athletic Group, Stage (of the exercise test), and Group*Stage interaction are shown (p, significance level; $\eta^2$, effect size).

phenylalanine, threonine, tryptophan), 7 non-essential (cystine, glutamine, glycine, proline, alanine, glutamic acid, and serine), and 8 non-proteinogenic (1-methyl-histidine, α-aminobutyric acid, β-alanine, ethanolamine, ornithine, phosphoethanolamine, sarcosine, and taurine) ones. In each case, the SMM-adjusted concentration was significantly higher in endurance- than sprint-trained athletes and the exercise-induced response was less pronounced ('flattened') in the latter group. Detailed data on individual PFAA concentrations are available in the S2 Table, S2 Fig.

## Discussion

### Main findings

We found that the level of resting PFAAs and the exercise-induced response is related to the type of metabolic adaptation. The concentrations of most PFAAs adjusted for SMM were consistently significantly higher in endurance- than sprint-trained athletes at rest, during exercise, and recovery. Importantly, the differences were observed for both non-proteinogenic and proteinogenic PFAAs. The response of virtually all PFAAs (except for β-alanine) to the acute progressive exercise until exhaustion was significant and specific for individual PFAAs or groups of them. The available research on healthy active or trained humans indicates that clear changes in proteinogenic PFAA concentrations during acute exercise are observed at or above intensities of ~70–75% $\dot{V}O_2$max [13, 21–30] with less pronounced changes below this threshold [8, 13, 23, 25, 30]. This study confirms these observations, but we observed for the first time that a similar principle applies to changes in non-proteinogenic PFAA concentrations and that the PFAA levels are sport-specific (see further sections for more comments). In addition, the exercise stage*group interaction effect was shown for some PFAAs (threonine, tryptophan, glutamine, glycine, proline, serine, 1-methylhistidine, α-aminobutyric acid, ethanolamine, ornithine, sarcosine, and combined non-proteinogenic PFAAs), meaning that the pattern of concentration changes differed between the groups. The difference generally

relied on a stronger exercise response in endurance athletes, i.e., more often significant differences between resting and subsequent exercise concentrations, whereas the responses in sprinters were more flattened (smaller or no differences between resting and exercise PFAA concentrations).

It should be noted that the groups studied had similar macronutrient intakes per kg body weight, with similar relative contributions to total energy intake. The percentage of energy intake from carbohydrate, protein, and fat was in accordance with dietary recommendations in sports, and the protein intake of 2.1–2.2 g·kg$^{-1}$ body weight was within the range of 1.7–2.2 g·kg$^{-1}$ body weight, suggested to be maintained in athletes engaged in intense training [31]. Also, the concentration of proteinogenic PFAAs in our athletes did not differ significantly from the levels observed in normally adapted high-performance athletes [32]. Thus, it appears that our athletes were not protein or amino acid deficient at the time of the study. Moreover, the average levels of resting creatine kinase, a common indirect measure of muscle damage, were within the range expected in competitive athletes [33] and did not differ significantly between endurance athletes and sprinters. Therefore, the groups appear equivalent regarding in macronutrient intake and physiological readiness for exercise testing, suggesting that significant differences in levels and exercise-induced responses of individual PFAAs or their groups were due to differences in sport-specific metabolic adaptations.

## Physiological background of differences

Different levels of PFAAs can be viewed as a manifestation of contrasting training adaptations and associated metabolic profiles. The exercise test used in this study was endurance in nature, posing a maximum challenge to metabolic aerobic mechanisms. To meet the high demands, many adaptations, factors, and mechanisms underlying chronic endurance exercise must function at the systemic, organ, tissue, and cellular levels [34–40]. They include muscle contractile protein, mitochondrial function, metabolic regulation, intracellular signaling, and transcriptional responses [36]. Increased skeletal muscle microvascular net size, better vasodilator/perfusion capacity, and improved oxygen extraction are of great importance [40]. Also, an increase in mitochondrial protein content is necessary [34]. Enzymes in β-oxidation, tricarboxylic acid (TCA) cycle, and electron transport system are elevated, accompanied by delayed increases in lactate concentration [39]. Muscle cells gain greater capacity to transport lactate across the sarcolemma, and muscle glycogen stores increase and are less rapidly depleted during submaximal exercise [35, 38]. Other adaptations and properties also promote high oxidative potential and resistance to fatigue [36, 37], e.g. a higher proportion of type I (oxidative) muscle fibers, which are characterized by higher capillary and mitochondrial density, activity of oxidative enzymes (citrate synthase, succinate dehydrogenase, 3-hydroxyl-CoA dehydrogenase), intramuscular triglyceride content, and content of specific monocarboxylate transporters enabling rapid transport of pyruvate, lactate, and ketone bodies across the cell plasma membrane. The groups tested in this study significantly differed in the level of maximal aerobic capacity ($\dot{V}O_2$max, Table 1), which can be taken as a global and synthetic indicator of the metabolic properties and mechanisms mentioned above. We assume, therefore, that the differences in PFAA levels are part of this complex picture.

The flow of PFAAs, both proteinogenic and non-proteinogenic, supports processes that are involved in energy metabolism (including but not limited to mitochondrial activity) and fatigue reduction. In the context of the availability (extraction and release) of circulating PFAAs for body tissues during exercise, especially working skeletal muscles, total blood and plasma volumes are also highly relevant. Expressed as mL per kg body mass, these volumes are 10–37% higher (in extreme cases exceeding 50%) in endurance than non-endurance athletes

[41–43], with the latter group not different from untrained individuals [43]. The significant differences between athletic groups persist after normalizing blood and plasma volumes per kg fat-free mass [42]. It is well-known that blood perfusion considerably increases in the active muscles, reaching 30–300 mL·min$^{-1}$·100 g SMM$^{-1}$ [44], and endurance-trained individuals have a clearly higher arteriolar and capillary density in skeletal muscle than sprint-trained ones [44, 45]. Taken together, this means that during exercise, the absolute amount of PFAAs (and other metabolites) transported by blood flowing through the working muscles per unit of time must be greater in endurance athletes, even at similar plasma concentrations. The expression of PFAA concentrations relative to SMM in our study reflects well the metabolic specificity of the sports groups studied and the differences between them, with endurance athletes possibly characterized by a more intensified amino acid turnover during progressive exercise.

### Proteinogenic PFAAs

The key metabolic pathways activated during acute aerobic exercise are the TCA cycle, glucose-alanine cycle, glutamine-glutamate cycle, urea cycle, purine nucleotide cycle, and aspartate-malate shuttle – in which proteinogenic AAs are, admittedly, not a direct fuel in energy metabolism, yet serve as indispensable substrates or links between those processes [for detailed review see: 3–6, 47]. In brief, virtually all proteinogenic AAs are involved in the TCA cycle by entering it as carbon skeletons in anaplerotic reactions to provide key intermediates (α-ketoglutarate, succinyl-CoA, fumarate, and oxaloacetate), or are converted into pyruvate and acetyl-CoA. Alanine aminotransferase reaction (pyruvate + glutamate → alanine + α-ketoglutarate) is the major anaplerotic reaction at the start of exercise (up to ~30 min), before glycogen stores are depleted and pyruvate availability for the TCA cycle is reduced. Thus, greater PFAAs availability, as for endurance athletes, promotes a higher concentration of TCA cycle intermediates, better maintenance of the TCA cycle flux, and more effective oxidation of acetyl-CoA.

Alanine and glutamine are the most abundant PFAAs synthesized *de novo* (from leucine, isoleucine, valine, glutamate, aspartate, and asparagine) in skeletal muscle proportionally to exercise intensity and released in large quantities into circulation [3–6, 46]. They are the basis for the elimination of amino groups from skeletal muscle in the form of non-toxic nitrogen carriers via glucose-alanine and glutamate-glutamine cycles. Consequently, muscle ammonia is finally transported to the liver and can be removed as urea via the urea cycle, in which ornithine, arginine, aspartate, and citrulline are the main reactants. The glucose-alanine cycle, in turn, is important for reconverting glucose from alanine in the liver. The purine nucleotide cycle, which regulates the levels of adenine nucleotides and is markedly active after exercise, requires aspartate and glutamate, with the latter used to neutralize the ammonia produced when adenosine monophosphate is converted into inosine monophosphate. Another cyclic metabolic pathway involving an amino acid is the aspartate-malate shuttle that transfers electrons generated during glycolysis across the inner mitochondrial membrane for oxidative phosphorylation. Also, arginine is the precursor of nitric oxide used by the vascular endothelium in vasodilation to increase blood flow. We assume that the differences related to the above mechanisms were behind the higher concentration of most proteinogenic PFAAs in the endurance than in the sprint group during acute progressive exercise.

### Non-proteinogenic PFAAs

The total absolute concentration of non-proteinogenic PFAAs was about 12-fold lower (on average ~220 μmol·L$^{-1}$ or ~6.3 μmol·L$^{-1}$·kg SMM$^{-1}$ for combined groups and test stages) compared to proteinogenic PFAAs (~2670 μmol·L$^{-1}$ or ~74 μmol·L$^{-1}$·kg SMM$^{-1}$). Nevertheless,

non-proteinogenic PFAAs also significantly responded to exercise, suggesting involvement in related metabolic processes. In one recent study [13], the response of six non-proteinogenic PFAAs to exercise was analyzed in physically active nonathletic young adults. The concentration of three PFAAs – ornithine, α-aminobutyric acid, and phosphoserine – significantly decreased at the end of exercise at the intensity of 75% $\dot{V}O_2$max (but not at 45% and 60%), with taurine, citrulline, and β-alanine showing no significant response at any intensity. In our study, only β-alanine concentration did not change significantly with exercise. The significant concentration change in most non-proteinogenic PFAAs in our study indicates that there is a minimum exercise intensity required to elicit a response, similar to what happens with proteinogenic PFAAs.

If one compares our specialized athletic groups, of particular interest may be the eight non-proteinogenic PFAAs that reached higher SMM-adjusted concentrations in endurance athletes than in sprinters, i.e. ornithine, β-alanine, ethanolamine, phosphoethanolamine, taurine, 1-methylhistidine, α-aminobutyric acid, and sarcosine. Plasma ornithine levels tended to decrease during progressive exercise and stabilized during recovery, suggesting its net uptake from blood circulation. The trend was significantly more pronounced in the endurance athletes. This amino acid is the central part of the urea cycle, thus it contributes to the disposal of excess toxic ammonia, the concentration of which is highest at the time of exhaustion. Exercising endurance athletes tend to have lower plasma ammonia concentration than sprinters [47]. Ornithine is part of the toxic ammonia elimination system together with alanine and glutamine. The higher levels of each of these three PFAAs in endurance athletes are thus indicative of higher activity of ammonia reduction pathways in this group.

Beta-alanine showed no significant changes with exercise, but its level was consistently higher in the endurance group. This amino acid is the rate-limiting precursor of carnosine, a dipeptide synthesized in human muscle tissue. Its main, but not the only, function is buffering intracellular $H^+$ ions derived from lactic acid produced during anaerobic glycolysis activated by high-intensity exercise. The muscle content of carnosine depends on its synthesis from histidine and β-alanine, the latter mainly produced in the liver and available in the extracellular environment via the circulatory system [48]. A markedly higher carnosine content is observed in glycolytic than in oxidative muscle fibers and, consequently, in sprint- than endurance-trained athletes, however, there is no evidence of elevated plasma carnosine levels [49]. Our results, i.e. the lack of detectable amounts of carnosine in plasma, may indicate that carnosine, even if released from muscle, is degraded already in the intercellular space and may only enter the bloodstream in the form of its metabolites. Considering the above, higher plasma concentration of β-alanine in endurance athletes may be determined by the greater endogenous liver synthesis or diminished release of muscle carnosine metabolites in sprint-adapted muscles.

Ethanolamine and its derivative phosphoethanolamine are precursors of the phospholipid phosphatidylethanolamine, which is typically found in the inner leaflet of the membranes of all cells, including muscle cells. Phosphatidylethanolamine is found in abundance in mitochondria and ethanolamine forms its head group. Free ethanolamine is also present at varying concentrations in bodily fluids [50]. Since ethanolamine plays a structural role in skeletal muscle cell membranes and is a breakdown product of phosphatidylethanolamine, it is suggested as an indicator of muscle turnover [51]. The role of phosphatidylethanolamine with its integral part ethanolamine includes, inter alia, stimulating oxidative phosphorylation activity. Importantly, research on cell lines revealed that the reduction of mitochondrial phosphatidylethanolamine impaired respiratory capacity, adenosine triphosphate production, and activities of electron transport chain complexes I and IV [52]. Therefore, the substantial increase in plasma ethanolamine and phosphoethanolamine concentrations with exercise intensity (presumably

due to the production and release from muscle) and their higher level in endurance than in sprint-trained athletes, as revealed in our study, can be attributed to specific differences in adaptation-related response to the increasing metabolic stress posed on aerobic metabolism.

Taurine is present in plasma, liver, kidney, and brain, but is especially abundant in skeletal muscle, with much higher concentrations in type I (oxidative) than type II (glycolytic) fibers. It plays a beneficial role in a variety of processes, including but not limited to energy metabolism. Its release into plasma is directly proportional to exercise intensity and is likely due to taurine role in the regulatory mechanism of calcium ions ($Ca^{2+}$) handling during muscle contraction [53]. Our endurance athletes had higher plasma taurine concentrations and its more pronounced exercise-induced response than sprint-trained ones, indicative of metabolic distinctiveness determined by specific metabolic adaptations.

Little is known about the exercise-related function of 1-methylhistidine (anserine derivative), α-aminobutyric acid (methionine, threonine, and serine catabolite), and sarcosine (involved in glycine metabolism). Their response to exercise and significantly higher concentration in endurance athletes suggests a metabolic role worthy of attention and further exploration.

The comparison of five non-proteinogenic PFAAs – citrulline, hydroxyproline, 3-methylhistidine, β-aminoisobutyric acid, and α-aminoadipic acid – did not reveal significant discrepancies in exercise-induced response between our endurance and sprint-trained athletes neither for absolute nor SMM-adjusted concentrations. The response of these amino acids to exercise in athletes has not yet been thoroughly analyzed. Citrulline is mainly produced in one of the central reactions in the urea cycle. Its potential role to improve skeletal muscle blood flow is suggested due to its effects on whole-body nitric oxide production and nitric oxide-dependent signaling pathways [5]. We did not find significant sport discipline-related differences in citrulline concentration, even though oxidative fiber-rich skeletal muscles (expected in greater quantity in our endurance athletes) have higher levels of citrulline than glycolytic fiber-rich ones [4]. The 'technical' reason for the lack of differences might be the very high inter-individual variability in citrulline levels, which blurred the emerging downward trend with exercise intensity in endurance athletes, while sprinters showed no noticeable change. Hydroxyproline is a derivative of proline, with almost all of the body's stores (~99.8%) found in collagen, and has an essential role in its stability [54]. Together with hydroxylysine, blood hydroxyproline level may be used as a measure of collagen breakdown after high-impact muscle-damaging bouts of plyometric exercise, however, reaching peak values only ~48 hours after exercise [55]. In our study, where a low-impact running exercise was used and a short recovery period analyzed, hydroxyproline levels decreased during exercise and stabilized afterward, therefore, presumably other phenomena played a role. 3-methylhistidine, being part of the muscle proteins actin and myosin, is considered a marker of myofibrillar proteolysis, despite some methodological reservations and drawbacks [4, 6, 7, 12]. The other two amino acids appear to play more of a health-related role. The L-form of β-aminoisobutyric acid is an interesting small-molecule myokine produced by skeletal muscle during exercise due to mitochondrial valine oxidation, acting on other tissues in an endocrine manner, e.g. on osteocytes, adipose tissue, or beta-oxidation of hepatic fatty acids. Its activity is inversely correlated with cardiometabolic risk factors, insulin resistance, and atherosclerosis via physical training [5, 14, 38]. Plasma α-aminoadipic acid (an intermediary biomarker of lysine and tryptophan metabolism) is strongly associated with the risk of developing diabetes and elevated for many years prior to the onset of this disease [56]. The clear lack of significant between-group differences in citrulline, hydroxyproline, 3-methylhistidine, β-aminoisobutyric acid, and α-aminoadipic acid suggests that their functioning and turnover during exercise is independent of the type of training-related metabolic adaptation in competitive athletes.

## Implications

Our study may be a suggestion to fine-tune dietary and supplementation recommendations for high-performance athletes depending on sports specialty. Current focus is on maximizing muscle protein synthesis after resistance exercise to maintain or increase SMM, while endurance athletes are recommended to achieve adequate carbohydrate intake with some protein addition to offset muscle (myofibrillar) damage and to support recovery and net protein balance [57, 58]. Conceivably, amino acid blends tailored to the metabolic requirements of specific exercise or training modalities could be beneficial in competitive sports. It is also worth considering non-proteinogenic amino acids, whose role in exercise has already been noted [59–61].

## Limitations and strengths

The results presented here only apply to high-performance athletes, not to the general population. Moreover, the potential implications refer only to males, as there are gender differences in the utilization of energy substrates, including amino acids [62], as well as in main factors limiting $\dot{V}O_2$max [46]. We used a short-term progressive exercise (duration 20–24 min) followed by a 30-minute recovery, so the revealed PFAA concentrations differed from those induced by prolonged constant-intensity exercise. In addition, the data does not provide information on the actual production and consumption of PFAAs, only possible to assess with more invasive techniques, such as biopsy or arteriovenous tracers – basically inapplicable in our study due to short exercise duration, frequent sampling, and unsteady blood flow. Dietary data were incomplete, but appear to be representative of the groups studied. Some disparity might have occurred at submaximal exercise intensities, as endurance athletes typically achieve a lower percentage of $\dot{V}O_2$max than sprinters at the same speed. Also, at the same speed, sprinters may be above, whereas endurance runners below the anaerobic threshold (not determined in this study). However, this does not change our interpretation, as the main group effects and interactions of the ANOVA analysis were calculated collectively for rest, successive exercise stages, and recovery.

The strengths and novelties of our study include (i) robust methods and procedures of PFAAs assay and analysis, (ii) direct comparison of PFAA responses to the same standard exercise until exhaustion between homogenous groups of highly trained humans representing opposite metabolic adaptations, (iii) a broad spectrum of PFAAs, especially the hitherto poorly studied non-protein PFAAs under exercise conditions, (iv) multiple blood sampling during exercise and recovery, which allowed us to more accurately track the course of changes in PFAA concentrations, and (v) expressing concentrations relative to SMM (the main recipient of blood flow during exercise) to more accurately depict the availability and turnover of circulating PFAAs.

## Conclusions

In conclusion, our study suggests that the magnitude of PFAAs response to incremental aerobic exercise until exhaustion and subsequent recovery is associated with the type of long-term metabolic adaptations associated with competitive sports participation. The obtained results may indirectly indicate a greater turnover and availability of circulating PFAAs for skeletal muscles and other body tissues in endurance- than in sprint-trained individuals during acute progressive exercise. Interestingly, concentrations of non-proteinogenic PFAAs, despite much lower plasma levels than the proteinogenic ones, also change in response to exercise loads and recovery, suggesting their potentially important, though less understood role in exercise metabolism.

## Supporting information

**S1 Table. Multiple reaction monitoring transitions and limits of quantitation.**
(PDF)

**S2 Table. Absolute and SMM-adjusted individual PFAA concentrations.**
(PDF)

**S1 Fig. Time course of individual PFAA concentrations (absolute values).**
(PDF)

**S2 Fig. Time course of individual PFAA concentrations (SMM-adjusted values).**
(PDF)

## Acknowledgments

We thank all the excellent athletes and their coaches for participating in the study.

## Author Contributions

**Conceptualization:** Krzysztof Kusy, Jacek Zieliński.

**Formal analysis:** Krzysztof Kusy.

**Funding acquisition:** Krzysztof Kusy.

**Investigation:** Krzysztof Kusy, Jan Matysiak, Zenon J. Kokot, Monika Ciekot-Sołtysiak, Agnieszka Klupczyńska-Gabryszak, Ewa Anna Zarębska, Szymon Plewa, Paweł Dereziński, Jacek Zieliński.

**Methodology:** Krzysztof Kusy, Jan Matysiak, Zenon J. Kokot, Jacek Zieliński.

**Project administration:** Krzysztof Kusy.

**Resources:** Krzysztof Kusy, Jan Matysiak, Zenon J. Kokot, Jacek Zieliński.

**Supervision:** Krzysztof Kusy.

**Validation:** Krzysztof Kusy, Jan Matysiak.

**Visualization:** Krzysztof Kusy.

**Writing – original draft:** Krzysztof Kusy.

**Writing – review & editing:** Krzysztof Kusy, Jan Matysiak, Zenon J. Kokot, Monika Ciekot-Sołtysiak, Agnieszka Klupczyńska-Gabryszak, Ewa Anna Zarębska, Szymon Plewa, Paweł Dereziński, Jacek Zieliński.

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
