## [Decision Letter · Decision Letter 0]

18 Jul 2024

PONE-D-24-00565Exercise-induced response of proteinogenic and non-proteinogenic plasma free amino acids is sport-specific: A comparison of sprint and endurance athletesPLOS ONE

Dear Dr. Kusy,

Thank you for submitting your manuscript to PLOS ONE. After careful consideration, we feel that it has merit but does not fully meet PLOS ONE’s publication criteria as it currently stands. Therefore, we invite you to submit a revised version of the manuscript that addresses the points raised during the review process.

We look forward to receiving your revised manuscript.

Kind regards,

Anil Bhatia, Ph.D

Academic Editor

PLOS ONE

Reviewers' comments:

Reviewer's Responses to Questions

**Comments to the Author**

1. Is the manuscript technically sound, and do the data support the conclusions?

Reviewer #1: Yes

Reviewer #2: Yes

Reviewer #3: Yes

2. Has the statistical analysis been performed appropriately and rigorously? 

Reviewer #1: Yes

Reviewer #2: Yes

Reviewer #3: Yes

3. Have the authors made all data underlying the findings in their manuscript fully available?

Reviewer #1: Yes

Reviewer #2: Yes

Reviewer #3: Yes

4. Is the manuscript presented in an intelligible fashion and written in standard English?

Reviewer #1: Yes

Reviewer #2: Yes

Reviewer #3: Yes

5. Review Comments to the Author

Reviewer #1: The manuscript is about comparison of exercise-induced response of proteinogenic and non-proteinogenic plasma free amino acids (PFAAs) between individuals with opposite metabolic adaptations to athletic training (sprint versus endurance athletes).

Although this topic was addressed relatively frequently in the 1990s, the authors of the present study applied a valuable methodological solution based on robust methods and procedures for PFAA determination and analysis (i.e., multiple blood draws and expression of concentrations relative to SMM). Unfortunately, the available literature mainly provides limited pre- and post-exercise data, ignoring the course of PFAA changes during progressive exercise.

Abbreviations and units

I would recommend writing the maximum oxygen uptake as „V̇O2max” not as „V̇O2max”. Please change throughout the text

When it refers to heart rate (HR) I would recommend using „b·min-1” rather than „bpm”. Please change throughout the text

Participants

Lines 76-78: Although the authors provide the inclusion criteria for the training experience of the participants (i.e. „involvement in a planned and structured training process for at least 5 years”), it is worthwhile to provide accurate data on the training age/experience of both groups. The authors can include this information in Table 1.

Study design

In general, the control of participants' diet is the biggest limitation of the study (especially protein supplementation), but for objective reasons it is very difficult to precisely control this issue within the implemented study model.

We know that environmental factors like diet (e.g. protein-rich diet in strength athletes), dietary supplement intake (e.g. amino acid supplements), medication and exercise training (e.g. muscle damage) should be recorded and ideally standardized as they can influence metabolite concentrations in blood.

At least an estimate of what the protein intake per kilogram of body weight was would have been useful. This is important in the context of the results obtained, because, for example chronic arginine aspartate supplementation in runners reduces total PFAA level at rest and during a marathon run. Moreover, the speed and strength training period strongly decreases the fasting concentrations of amino acids in the power-trained athletes in a good anabolic state with the daily protein intake of 1.26 g·kg-1 body weight. At the same time the intensive lactic exercise session induces strong decreases, especially in valine, asparagine, and taurine.

Practically, on the day before testing, protein intake should have been standardized to 20% of total macronutrient intake.

Lines 100-101: The authors use a vague description of the PFAA analysis, writing that they used a "well-established and validated method". It is advisable to include a detailed description of the method used.

Statistical analysis

Lines 200-210: Can the authors provide the name of the statistical program with which they performed the statistical analyses (e.g. Statistica v13.3 PL, IBM® SPSS)?

Results

The results are a very strong part of the manuscript, they were prepared with precision, contain in-depth analysis and look very good graphically.

Future research

If the authors plan further research in this area, it would be worthwhile to study the interactions between genotype and metabolome.

Influences on the metabolome are manifold and further studies are needed to disentangle the specific contributions of genetic variants, of adaptations to sports-specific exercise training or of special nutrition to the systematic metabolic differences between differently trained individuals. Genetics contribute to metabolite and protein concentration differences between subjects and can be responsible for up to 75% (for metabolites) of the metabolite concentration in blood. Especially in small and highly specialized cohorts, genetics can influence outcomes in exercise studies.

Reviewer #2: The reviewer’s comments on the paper are as follows:

I have reviewed the research type article entitled “Exercise-induced response of proteinogenic and non-proteinogenic plasma free amino acids is sport-specific: A comparison of sprint and endurance athletes”.

It is interesting topic and totally, this manuscript is well organized and written. However, there are just a few points of clarification that need to be addresses. Therefore, I recommend that the manuscript must be revised before it can be reconsidered for publication. Here are my comments:

- Page 3, Introduction, Line 44-45: The authors wrote “Circulating blood plasma is a temporary and small but important reservoir of free AAs (~1% of total AAs)”. If plasma is temporary tank of FAAs and including one percent of total body AAs, please explain why PFAAs could have crucial role regard to exercise metabolism in particular sports specialization?

- Page 5, Line 107: Considering possible effects of diet especially day before testing, please describe more details of a 48-hour recall of participants’ dietary intake consist of total daily calorie intake, macronutrients distribution, and percent protein consumption. These data should be reported as a separate table.

- Page 5, Exercise protocol, Line 125-126: It has been reported that total exercise duration ranged from 20 min (sprinters) to 24 min (endurance athletes). This point can be important at the observed changes in variables (Lactate, PFAAs, etc.) between two kinds of exercise metabolism and so their interpretation. Therefore, please include this issue for relevant discussion.

- Please clarify why the researchers choose incremental running exercise as identical test for investigating the response of participants? What’s the reasons for? Why two tests weren’t selected fit to both sport events corresponding to specific adaptations?

- Blood samples were drawn 11 times before, during and after exercise test. 5 of them were in time of doing exercise. Weren’t these sampling effective on participants’ performance? If so, please mention it as limitation.

- I didn’t see an explanation about participants’ sleep. Did the researchers give advices to them? Please clarify.

- Page 5, Line 104: The authors wrote “Athletes arrived at the laboratory in the morning after an overnight fast”. What time the test was conducted exactly? Please write.

- Page 7, Other blood assays, Line 193-197: Please mention sensitivity or CV of measurements for blood lactate, ammonia, and creatine kinase.

- Page 8, Table 1: Fat mass percent of endurance and sprint athletes were 13.7 vs. 10.7 respectively. Are those correct? Please just recheck.

- Page 15, Implications, Line 429-455: It’s a bit long. Please summarize content without repeating the literature. Moreover, considering a lot of limitations in this study rewrite some sentences (practical messages) with possibility and uncertainty.

- Page 17, Conclusions, Line 475-481: This sub-section need to be revised on basis of previous comment. Caution should be considered when writing the concluding sentences in particular the first sentences.

Regards,

Reviewer #3: The publication titled “Exercise-induced response of proteinogenic and non-proteinogenic plasma free amino acids is sport-specific: A comparison of sprint and endurance athletes” is a significant study examining how exercise affect the levels of proteinogenic and non-proteinogenic plasma free amino acids. I recommend that the editor to accept this manuscript after a minor revision addressing the following concerns:

1. The supporting information is missing from the original file. The author should upload the supporting information when submitting the revision.

2. The author should consider arranging the figures horizontally instead of vertically.

6. PLOS authors have the option to publish the peer review history of their article (what does this mean?). If published, this will include your full peer review and any attached files.

Reviewer #1: No

Reviewer #2: No

Reviewer #3: No

---

## [Author Response · Author response to Decision Letter 0]

1 Aug 2024

Reviewer #1: 

The manuscript is about comparison of exercise-induced response of proteinogenic and non-proteinogenic plasma free amino acids (PFAAs) between individuals with opposite metabolic adaptations to athletic training (sprint versus endurance athletes).

Although this topic was addressed relatively frequently in the 1990s, the authors of the present study applied a valuable methodological solution based on robust methods and procedures for PFAA determination and analysis (i.e., multiple blood draws and expression of concentrations relative to SMM). Unfortunately, the available literature mainly provides limited pre- and post-exercise data, ignoring the course of PFAA changes during progressive exercise.

Thank you for your overall positive assessment. We realize, there are many studies dealing with various aspects of exercise-induced amino acid metabolism. We are trying to complete the picture by analyzing proteinogenic and non-proteinogenic amino acids at the same time, taking blood samples more frequently, and, most importantly, directly comparing in a single study individuals with extremely different metabolic adaptations (such attempts have not been made to our knowledge). 

Abbreviations and units

I would recommend writing the maximum oxygen uptake as „V̇O2max” not as „V̇O2max”. Please change throughout the text.

Thank you for your comment. There are simplified versions of the abbreviation for maximal oxygen consumption used in various scientific and popular texts. We are not sure about the Reviewer’s intention, as both abbreviations in the above comment are written identically. Dot above V is obvious (volumetric flow) and this is the notation we used. The ‘2’ should be and is in subscript, so we think it is about "max", which we in the manuscript write as subscript. In fact, writing "max" in the normal (base)line of type may be more correct, so we changed the subscript “max” to normal line. We have also consistently corrected the abbreviations for HRmax, NH3max, and LAmax. 

When it refers to heart rate (HR) I would recommend using „b·min-1” rather than „bpm”. Please change throughout the text

Thank you. Corrected.

Participants

Lines 76-78: Although the authors provide the inclusion criteria for the training experience of the participants (i.e. „involvement in a planned and structured training process for at least 5 years”), it is worthwhile to provide accurate data on the training age/experience of both groups. The authors can include this information in Table 1.

Thank you for indicating this detail. The data have been supplemented in Table 1 as years of ‘specialized training’.

Study design

In general, the control of participants' diet is the biggest limitation of the study (especially protein supplementation), but for objective reasons it is very difficult to precisely control this issue within the implemented study model.

We know that environmental factors like diet (e.g. protein-rich diet in strength athletes), dietary supplement intake (e.g. amino acid supplements), medication and exercise training (e.g. muscle damage) should be recorded and ideally standardized as they can influence metabolite concentrations in blood.

At least an estimate of what the protein intake per kilogram of body weight was would have been useful. This is important in the context of the results obtained, because, for example chronic arginine aspartate supplementation in runners reduces total PFAA level at rest and during a marathon run. Moreover, the speed and strength training period strongly decreases the fasting concentrations of amino acids in the power-trained athletes in a good anabolic state with the daily protein intake of 1.26 g·kg-1 body weight. At the same time the intensive lactic exercise session induces strong decreases, especially in valine, asparagine, and taurine.

Practically, on the day before testing, protein intake should have been standardized to 20% of total macronutrient intake.

Thank you for this crucial comment. We fully agree on the relevance of the factors mentioned by the reviewer that affect the concentration of amino acids in the blood. In our study, we can exclude the influence of medication, since the athletes studied were healthy and were not treated for any chronic or acute diseases. Sports training specificity was, by design, radically different in the two groups (that's why we compared them), thus the "standardization" of training could only consist in the cessation of strenuous high-intensity and/or long-duration training sessions 24‒48 hours before the laboratory visit in order to keep the athletes sufficiently rested and recovered (to which the coaches and athletes willingly adapted, since they were motivated for obtaining the best possible test performance and diagnose their cardiorespiratory and body composition parameters to support training control and planning). Please note that the average creatine kinase levels (CK; a common indirect measure of muscle damage, despite some controversies and areas to be clarified) on the day of exercise testing, provided in table 1, are relatively low in our athletes, considering that higher CK levels/reference values are expected in athletes than in general population (see e.g. 10.1136/bjsm.2006.034041). Importantly, CK levels did not differ significantly between endurance athletes and sprinters. Therefore, we believe that we have managed to control the muscle damage factor to a sufficient extent.

As for diet and dietary supplements control, we collected such data (estimates of energy, protein, fat, and carbohydrate intake) and the new manuscript has been updated accordingly (methods, results, discussion). The limitation is that we did not obtain complete records from all athletes (7/11 in the endurance group and 6/10 in sprinters). That is why we did not include this data in the original version. However, we currently believe that even partial data can be representative and valuable for better interpretation of the results. It is most important that the studied (sub)groups did not differ significantly in the consumption of macronutrients per kg of body weight and the percentage contribution in the total energy intake. Thus, both sports groups seem to be equivalent ('standardized') in this respect, which strongly suggests that the clearly different levels and exercise-induced responses of certain plasma amino acids are due to differences in sport-specific metabolic adaptations. Overall, the percentage of energy intake from carbohydrate, protein and fat was in accordance with dietary recommendations of the Int. Soc. Sports Nutr., and the protein intake per kg body weight (2.1‒2.2 g/kg body weight) was that expected in athletes engaged in intense training (1.7‒2.2 g/kg body weight; Kersick et al. 2018, doi: 10.1186/s12970-018-0242-y). In addition, the concentration of proteinogenic plasma amino acids in our athletes did not differ significantly from the ranges shown in normally adapted high-performance athletes (Kingsbury et al. 1998, doi: 10.1136/bjsm.32.1.25). Thus, it appears that our athletes were not protein or amino acid deficient at the time of the study.

As for the day before testing, athletes were asked not to modify their dietary habits and not to hydrate extra or dehydrate before the test, not to intake supplements or aids potentially affecting amino acid levels and physical performance. To check this, they reported on their food and supplementary intake over the past 48 hours to detect unusual eating behaviors that could affect the study results (in which case laboratory measurements could be rescheduled). The energy contribution from proteins was on average within a range of 16‒17% in our athletes, thus within the typical/normal range as shown in Table 2 (Kersick et al. 2018, doi: 10.1186/s12970-018-0242-y). We agree that a standardizing to 20% could be more precise, but on the other hand, it is unclear how a sudden one-time change in dietary routine would affect the overall metabolic state in the athletes.

Lines 100-101: The authors use a vague description of the PFAA analysis, writing that they used a "well-established and validated method". It is advisable to include a detailed description of the method used.

Thank you for this comment. This section contains only main assumptions of the study design, hence it is a general description. To avoid confusion, the part of the sentence that reads “using a well-established and validated method” has been removed. The details of the analytical method with further references are described in the section “Plasma free amino acids assay”.

Statistical analysis

Lines 200-210: Can the authors provide the name of the statistical program with which they performed the statistical analyses (e.g. Statistica v13.3 PL, IBM® SPSS)?

Thank you for noticing this. All statistical analyses were performed using the Statistica 13.3 software package (TIBCO software Inc., Santa Clara, CA, USA). The information has been supplemented.

Results

The results are a very strong part of the manuscript, they were prepared with precision, contain in-depth analysis and look very good graphically.

Thank you very much for your positive evaluation. We tried to do our best.

Future research

If the authors plan further research in this area, it would be worthwhile to study the interactions between genotype and metabolome.

Influences on the metabolome are manifold and further studies are needed to disentangle the specific contributions of genetic variants, of adaptations to sports-specific exercise training or of special nutrition to the systematic metabolic differences between differently trained individuals. Genetics contribute to metabolite and protein concentration differences between subjects and can be responsible for up to 75% (for metabolites) of the metabolite concentration in blood. Especially in small and highly specialized cohorts, genetics can influence outcomes in exercise studies.

It is hard not to agree with the reviewer. A more comprehensive combined metabolomic and genomic approach could help better interpret the exercise responses, especially in the case of athletes, who in principle must have certain genetic predispositions to perform at the highest level in sports. The contribution of genetic and environmental (training) factors to exercise response, physical fitness and sports performance is still a topical issue. So far, our team has made one modest attempt in a related topic (https://doi.org/10.1007/s13353-022-00736-6).

Reviewer #2: The reviewer’s comments on the paper are as follows:

I have reviewed the research type article entitled “Exercise-induced response of proteinogenic and non-proteinogenic plasma free amino acids is sport-specific: A comparison of sprint and endurance athletes”.

It is interesting topic and totally, this manuscript is well organized and written. However, there are just a few points of clarification that need to be addresses. Therefore, I recommend that the manuscript must be revised before it can be reconsidered for publication. Here are my comments:

Thank you very much for your overall positive assessment. Below, we address the valuable comments. Appropriate changes were also made to the manuscript.

- Page 3, Introduction, Line 44-45: The authors wrote “Circulating blood plasma is a temporary and small but important reservoir of free AAs (~1% of total AAs)”. If plasma is temporary tank of FAAs and including one percent of total body AAs, please explain why PFAAs could have crucial role regard to exercise metabolism in particular sports specialization?

Plasma amino acid stores are temporary in the sense that their levels can change rapidly due to various factors, and in our study such a factor is exercise of increasing intensity. But, of course, the flow of this fraction of the amino acid pool in the circulatory system is continuous and ceaseless. It is well known that with exercise intensity, blood flow in skeletal muscles increases dramatically, and with it the rate of amino acid transport flow. During exercise, the circulatory system is essentially the only fast and efficient route for the exchange of amino acids between muscles and other tissues and organs. It allows the efficiency of metabolic processes to be continuously adjusted to meet the demands of physical exertion. The plasma PFAA concentration profile can be considered a good indirect indicator of amino acid turnover during exercise and the "strategy" of their utilization depending on the sports specialty. A relevant brief explanation is included in the introduction.

As an aside, it can be added that typically only the 'static' concentration of amino acids at a given exercise intensity is measured. It would be even more interesting to measure the magnitude and changes in PFAAs flow, e.g. in micromoles per liter per minute (or production and consumption of PFAAs as mentioned in the limitation section). Unfortunately, more invasive methods, not feasible for some reasons, would have to be used.

- Page 5, Line 107: Considering possible effects of diet especially day before testing, please describe more details of a 48-hour recall of participants’ dietary intake consist of total daily calorie intake, macronutrients distribution, and percent protein consumption. These data should be reported as a separate table.

Thank you for this crucial comment. As for diet and dietary supplements control, we had collected such data (estimates of energy, protein, fat, and carbohydrate intake) and the manuscript has been updated accordingly (methods, additional table in the results section, discussion). The limitation is that we did not obtain complete records from all athletes (7/10 in the endurance group and 6/10 in sprinters). That is why we didn't include this data in the original version. However, we currently believe that even partial data can be representative and valuable for better interpretation of the results. It is most important that the studied (sub)groups did not differ significantly in the consumption of macronutrients per kg of body weight and the percentage contribution in the total energy intake. Thus, both sports groups seem to be equivalent ('standardized') in this respect, which strongly suggests that the clearly different levels and exercise-induced responses of certain plasma amino acids are due to differences in sport-specific metabolic adaptations. Overall, the percentage of energy intake from carbohydrate, protein and fat was in accordance with Int. Soc. Sports Nutr. (ISSN) recommendations, and the protein intake per kg body weight (2.1‒2.2 g/kg body weight) was typical of athletes engaged in intense training (1.7‒2.2 g/kg body weight; Kersick et al. 2018, doi: 10.1186/s12970-018-0242-y). In addition, the concentration of proteinogenic plasma amino acids in our athletes did not differ significantly from the ranges shown in normally adapted high-performance athletes (Kingsbury et al. 1998, doi: 10.1136/bjsm.32.1.25). Thus, it appears that our athletes were not protein or amino acid deficient at the time of the study. The manuscript has been updated accordingly (methods, results, discussion).

By the way, the 48-hour recall of participants' dietary intake was just an additional quick check to detect abnormal 'momentary' eating behavior. 

- Page 5, Exercise protocol, Line 125-126: It has been reported that total exercise duration ranged from 20 min (sprinters) to 24 min (endurance athletes). This point can be important at the observed changes in variables (Lactate, PFAAs, etc.) between two kinds of exercise metabolism and so their interpretation. Therefore, please include this issue for relevant discussion.

Thank you for raising this important question. Obviously, with an identical exercise protocol until exhaustion (same speed progression), our endurance athletes reached higher running speeds, and the test lasted longer for them. However, in the end, at the point of exhaustion, athletes from both groups presented values of physiological and biochemical indicators that objectively reflected their maximal effort/fatigue and subsequent recovery. In this regard, the groups are fully comparable (including heart rate, oxygen uptake, lactate, ammonia and all PFAAs). In this sense, the d

---

## [Decision Letter · Decision Letter 1]

14 Aug 2024

Exercise-induced response of proteinogenic and non-proteinogenic plasma free amino acids is sport-specific: A comparison of sprint and endurance athletes

PONE-D-24-00565R1

Dear Dr. Kusy,

We’re pleased to inform you that your manuscript has been judged scientifically suitable for publication and will be formally accepted for publication once it meets all outstanding technical requirements.

Kind regards,

Anil Bhatia, Ph.D

Academic Editor

PLOS ONE

Additional Editor Comments (optional):

Reviewers' comments:

Reviewer's Responses to Questions

**Comments to the Author**

1. If the authors have adequately addressed your comments raised in a previous round of review and you feel that this manuscript is now acceptable for publication, you may indicate that here to bypass the “Comments to the Author” section, enter your conflict of interest statement in the “Confidential to Editor” section, and submit your "Accept" recommendation.

Reviewer #1: All comments have been addressed

Reviewer #2: All comments have been addressed

Reviewer #3: All comments have been addressed

2. Is the manuscript technically sound, and do the data support the conclusions?

Reviewer #1: Yes

Reviewer #2: Yes

Reviewer #3: Yes

3. Has the statistical analysis been performed appropriately and rigorously? 

Reviewer #1: Yes

Reviewer #2: Yes

Reviewer #3: Yes

4. Have the authors made all data underlying the findings in their manuscript fully available?

Reviewer #1: Yes

Reviewer #2: Yes

Reviewer #3: Yes

5. Is the manuscript presented in an intelligible fashion and written in standard English?

Reviewer #1: Yes

Reviewer #2: Yes

Reviewer #3: Yes

6. Review Comments to the Author

Reviewer #1: Thank you for your detailed responses to my comments and the appropriate changes to the text. The work has a lot of potential, so I encourage you to continue the research topic undertaken.

Reviewer #2: Thanks and congratulations to all authors of the paper. The authors responded to the comments satisfactorily. Therefore, the current form of the manuscript appears appropriate for publication in PLOS ONE.

Regards,

Reviewer #3: The authors have addressed all the concerns. I recommend the editor except this manuscript without further revision.

7. PLOS authors have the option to publish the peer review history of their article (what does this mean?). If published, this will include your full peer review and any attached files.

Reviewer #1: No

Reviewer #2: **Yes: **Hamid Arazi

Reviewer #3: No

---

## [Editor Report · Acceptance letter]

22 Aug 2024

PONE-D-24-00565R1 

PLOS ONE

Dear Dr. Kusy, 

I'm pleased to inform you that your manuscript has been deemed suitable for publication in PLOS ONE. Congratulations! Your manuscript is now being handed over to our production team.

Kind regards, 

on behalf of

Dr. Anil Bhatia 

Academic Editor

PLOS ONE